# Qualitative Interview Study of Gynecologic Oncologist Utilization of Recommended Same-Day Discharge Following Minimally Invasive Hysterectomy

**DOI:** 10.3390/jpm12071082

**Published:** 2022-06-30

**Authors:** Sophia Bunde, Shalkar Adambekov, Ella Glikson, Faina Linkov

**Affiliations:** 1Department of Obstetrics, Gynecology, and Reproductive Sciences, University of Pittsburgh School of Medicine, Pittsburgh, PA 15213, USA; faina.linkov@gmail.com; 2Department of Epidemiology, Biostatistics, and Evidence Based Medicine, Al-Farabi Kazakh National University, Almaty 05004, Kazakhstan; SHA70@pitt.edu; 3The Graduate School of Business Administration, Bar Ilan University, Ramat Gan 5290002, Israel; ella.glikson@gmail.com

**Keywords:** same-day discharge, hysterectomy, gynecologic oncology, value of care, minimally invasive surgery

## Abstract

Recent investigations have supported the safety and benefits of discharging women on the same day following a minimally invasive hysterectomy (MIH) for both benign and malignant indications. Not all eligible candidates for same-day discharge (SDD) are discharged the same day, and patients undergoing an MIH for malignant indications have decreased the odds of receiving SDD despite established safety. The objective of this study was to use qualitative interviews to explore physician decision making regarding SDD after an MIH for malignant indications. Six qualitative interviews of gynecologic oncologists were analyzed using recurrent theme analysis for distinct themes in physician decision making regarding SDD. Results suggest that physician-perceived barriers to SDD include patient health characteristics, patient social characteristics, and hospital-system factors. Cited factors influencing SDD include patient travel, social support, practice setting (urban vs. rural) and staff comfort with the recommendation. Obstructive sleep apnea and post-surgical oxygenation appear to be a recurring reason for unplanned admission. The utilization of SDD after an MIH in the gynecologic oncology patient population is influenced by patient, physician, and system factors. Addressing the physician’s perceived barriers to SDD and catering recommendations to the gynecologic oncology population may increase utilization.

## 1. Introduction

A hysterectomy, the surgical removal of the uterus, is the second most common surgical procedure in US women, following a cesarean section. A hysterectomy is performed for both benign and oncologic indications [1]. Certain types of ovarian, endometrial, and cervical cancers involve a hysterectomy with or without the concurrent removal of ovaries, fallopian tubes, surrounding connective tissue, and lymph node sampling or complete dissection as part of the staging and treatment. Recently, minimally invasive hysterectomy (MIH) techniques have replaced the total abdominal hysterectomy (TAH) as the default approach in appropriate patients. Vaginal, laparoscopic, laparoscopic-assisted vaginal, and robotic-assisted hysterectomy are all considered MIH techniques and are preferred to TAH for most patients [2]. Despite the additional complexity of a hysterectomy in oncology, many patients are good candidates for an MIH unless their uterine size, advanced disease, or pelvic outlet obstruction make it logistically difficult [3].

Same-day discharge (SDD) has been extensively studied as a safe and feasible option for women undergoing an MIH for both oncologic indications, as well as abnormal uterine bleeding, pelvic organ prolapses, fibroids, and other benign indications. With the introduction of MIHs, what used to be a major abdominal surgery became an outpatient procedure. SDD has been shown to yield reduced rates of readmission, reoperation, and post-operative complications compared to overnight admissions [4,5,6,7,8]. Shortened hospital stays are associated with decreases in infection, costs, and time to recovery, while maintaining equivalent rates of post-operative readmission [9,10,11].

As SDD has become the standard of care, research has revealed factors associated with the reduced odds of SDD, including an older patient age, higher BMI, and increased complexity of the MIH operation [4,7,12]. The oncology population tends to be older and less healthy than the general gynecology population; though patient demographics are different between types of gynecologic malignancies [13]. However, investigations into post-operative complications related to these factors have not shown worse post-operative outcomes [4,6]. Malignancy itself is associated with lower rates of SDD despite evidence that admission does not change post-surgical outcomes [4,6,14]. Specifically, advanced age and length of surgery in gynecologic oncology make SDD less likely [15]. The analysis of practice data of a large academic center indicates that patients undergoing MIH for malignant indications receive SDD at a lower rate than other patients, and gynecologic oncologists admit more patients following MIH than their non-oncologic peers [16].

Thus, the factors identified as related to low SDD do not appear to be clearly associated with increased post-surgical needs prompting admission. The factors influencing physicians’ decisions to utilize SDD following MIH remain unclear. There is, however, evidence that factors outside of patients’ objective clinical stability are involved in decision making [4]. A better understanding of this decision making may provide direction in efforts to increase SDD among gynecologic oncology patients, increasing care quality by maintaining and improving patient safety and outcomes while optimizing costs.

Qualitative research provides an important tool to improve the understanding of the complexity of physician utilization of SDD and works to bring together quantitative evidence with the reality of clinical practice [17,18]. Qualitative interviews allow for exploring subtleties of research questions that cannot be captured by quantitative investigations—such as physician and patient attitudes toward recommended clinical practice pathways, as well as physician decision-making processes in following or not following such recommendations. While our group has previously investigated the complication rates, lengths of stay, variable uptake of the MIH and SDD pathway amongst subspecialists, and patient racial and socioeconomic factors associated with MIH use at our institution, there are remaining gaps in the knowledge of barriers to SDD utilization [4,16,19,20,21,22]. Therefore, the key aim of this qualitative research is to expand the current understanding of the factors influencing SDD following an MIH in oncological practice from a physician’s perspective.

## 2. Materials and Methods

### 2.1. Data Collection

This is a qualitative research study aiming to understand current barriers to SDD following MIH by targeting physician perception of existing decision processes. Following practice trends, the UPMC Health System (UPMC) implemented a clinical pathway to increase physician utilization of MIH techniques in 2012. UPMC is a large health system with urban, suburban, and rural hospitals in Pennsylvania. This pathway successfully increased utilization of MIH without increasing postoperative complications [3,18,19,20,21]. In 2014, UPMC added a recommendation of SDD to the hysterectomy pathway.

### 2.2. Study Participants

Six physicians from The Division of Gynecologic Oncology within the Department of Obstetrics, Gynecology, and Reproductive Sciences, UPMC, were interviewed. These physicians practice primarily at the large tertiary care center, but additionally operate at suburban and rural hospitals within the UPMC Health System. The study was deemed to be IRB exempt on the basis that the interviewees were not sharing protected health information, but were rather participating in their role as physicians (University of Pittsburgh IRB ID Study19020316). Email requests for participation were sent directly to physicians within the division. The characteristics of the patients that these physicians serve have been described in our previous publications, with the mean age of these patients being at 48 years [22].

### 2.3. Data Collection

Interviews occurred in private offices. After reviewing the written information sheet and verbally consenting to participate, physicians were asked questions regarding factors that impact their decision to admit or discharge a patient after MIH (Table 1), and given a brief written survey to collect demographic information. The interviews were semi-structured: each physician was asked these questions, which were followed up with clarifying questions as needed. The interviews were conducted and audio-recorded by a single investigator and securely stored.

The interviewer transcribed audio recordings verbatim, and any identifying information was removed. The software program NVivo (12.2.0) was used to code and analyze the transcripts. The transcripts were reviewed for themes in physician decision-making regarding SDD. Instead of a pre-set codebook, the codebook was created via a synchronous process, per Crabtree and Miller’s editing style of qualitative research [23]. The codebook was continuously updated with each interview, with previous interviews recoded with each iteration of the codebook until interviewing ceased.

### 2.4. Data Analysis

The coded interviews were analyzed using recurrent theme analysis. Related ideas were aggregated into meta-themes and analyzed for frequency across interviews. Beyond the frequency by which certain themes appeared, the coded transcripts were analyzed for thematic connections. Relationships between certain aspects of the physician answers were examined for patterns; for example, a notable pattern of the code [“concern for patient support” occurring with relation to the code “physician discussing pre-operative counseling”] emerged. Consistent with best practices for qualitative research, data saturation was used to determine the sample size. In this study, data saturation occurred in around 4–5 interviewed physicians, meaning that the study reached a point where there were no new data obtained from additionally interviewed physicians [24].

## 3. Results

Six gynecologic oncology physicians from The Department of Obstetrics, Gynecology and Reproductive Sciences were interviewed in person over a period of six months. The average age of the physicians was 42 years, and the average time since fellowship training was 8.16 years. Physicians interviewed in this study perform procedures for both oncological and benign indications.

Recurrent theme analysis revealed that barriers to SDD fall into specific categories: patient factors, physician factors, and systems factors (Figure 1). Recurring themes of patient barriers to SDD reveal that there is a physician perception of general poor surgical candidacy among patients based on advanced age and a high incidence of comorbid conditions. Additionally, malignancy itself and the resultant need for prompt surgery were cited consistently as a reason for the low utilization of SDD.

Physician-related barriers included the physicians’ assessment of patient frailty and their ability to effectively counsel on SDD. Physicians revealed discomfort with SDD if it resulted in patients feeling uncared for. Themes that emerged in systemic barriers to SDD focused on the support of the recommendation by nursing staff, the difference in post-surgical standard procedure among hospital sites, and the slow adaptation to a change in protocol since the implementation of the recommendation.

### 3.1. Obstructive Sleep Apnea

Physicians reported a recurrent issue of post-surgical oxygenation for patients with obstructive sleep apnea (Table 2). Difficulty in weaning patients off oxygen therapy post-operatively was discussed as a major reason for admitting patients despite an initial plan of SDD. Specifically, the physicians cited patients with poorly managed obstructive sleep apnea (OSA) or patients with currently undiagnosed OSA as particularly problematic.

More generally, physicians emphasized that medical comorbidities make the oncology patient population more medically complex than those undergoing a hysterectomy for benign indications (Table 3).

### 3.2. Hospital Site Difference

The physicians interviewed were selected within the UPMC hospital system, which allowed for the participation of doctors that practice in different hospital locations throughout the metropolitan Pittsburgh area and the surrounding counties. The size and geographic reach of UPMC extends urban, high-volume hospital policy into rural hospitals, creating a complex intersection of different hospital cultures, staffs, and patient populations that physicians suggested affect adherence to the SDD recommendations. Physicians that perform MIH at multiple sites within the hospital system described different practical application of SDD between sites (Table 4).

### 3.3. Travel Time to Hospital

The physicians discussed the difficulty of discharge when patients had traveled significant distances for care and underlined the importance of preoperative planning in order to achieve SDD (Table 5). Importantly, physicians reported that issues of distance are compounded when patients lack social support or financial resources, and thus patients are sometimes admitted because of transportation barriers rather than medical necessity. One physician explained that the matter is further complicated when patients wish to stay overnight—with both patient satisfaction and departmental goals of SDD tied to physician reimbursement, adherence to the recommendation of SDD becomes increasingly complex.

## 4. Discussion

### 4.1. Statement of Principal Findings

In this study, gynecologic oncology providers identify unique barriers to SDD within the oncology population that are not currently addressed. These barriers, including the inability to optimize patients due to the need for timely operation, patients’ travel distance, access to resources, and support to navigate this travel, OSA and resultant post-operative issues, and inconsistency in the adoption of the recommendation between hospital sites. Overall, the physicians cited that advanced age and the increased incidence of medical comorbidities made SDD less feasible in their patient population as compared to MIH in benign gynecology.

### 4.2. Interpretation within the Context of the Wider Literature

The importance of OSA within gynecologic oncology populations has been established, with studies reporting rates of OSA between 30 and 50%, much higher than among the general adult surgical populations and hypothesized to be related to the prevalence of obesity in this population [25,26]. Several reviews have examined the safety of ambulatory surgery for patients with a diagnosis or high suspicion of obstructive sleep apnea. While there appears to be no higher risk of hospital readmission for patients with OSA, there are more immediate post-surgical respiratory events in this population [26,27]. Patients with undiagnosed OSA are found to have higher risk of post-surgical adverse outcomes than patients previously diagnosed, suggesting the need for early identification and treatment of OSA to allow for prompt discharge [28]. In a prospective observational study of 383 surgical patients in the gynecologic oncology population, patients that screened positive via a STOP questionnaire and those with sleep oximetry consistent with OSA (38% of patients) were more likely to have post-operative hypoxemia [26].

These investigations align with physician perspectives of barriers to SDD in patients with OSA, suggesting that SDD is more difficult in this population, particularly without pre-surgical screening and treatment. Similarly, in a survey of French gynecologic surgeons, 77% of responding surgeons rated no history of OSA as the criteria for appropriateness of an outpatient hysterectomy, corroborating this study’s finding of OSA as a perceived barrier to successful SDD [29]. As referenced by the physicians, patients that are non-adherent to therapy pose a similar post-surgical risk of needing additional oxygen therapy to those undiagnosed at the time of surgery.

The physicians described the additional consideration of patient travel when deciding on the discharge course following an MIH. The issue of patient travel in oncology care has been examined in various cancer populations; patients in rural areas routinely travel far beyond local health care centers for oncology care and 14.8 million women reside more than 50 miles from gynecologic oncologic care [30,31]. A qualitative interview study recruited 19 women that traveled an average of 87 miles to receive gynecologic oncology care. The study revealed themes of timing and the need to coordinate a companion as barriers to receiving care and reported that most patients had to utilize personal resources or utilize community programs (similar to Family House at UPMC) in order to participate in non-surgical care away from home [32]. This study corroborated the perspective of interviewed physicians that social support was essential for patients that had to travel for oncologic care. Therefore, in order to practically address SDD in oncology care, attention must be paid to the logistic factors affecting large portions of the patient population.

The interviewed providers perform MIHs at multiple hospital sites within the UPMC Health System and described disparate times to discharge between hospital sites despite the universal Enhanced Recovery After Surgery (ERAS) protocol and recommendation for SDD. This suggests that factors beyond the recommendation and protocol affect discharge times. Length-of-stay has been shown to be shorter in higher-volume institutions, and implementing system-wide practice algorithms works best when training is tailored to specific hospital settings within the system [33,34]. While clinical protocols work to standardize care, effective implementation must take into consideration current practice protocols and culture.

Physician education has shown to be a more influential factor in the increased utilization of SDD than patient preparation, suggesting that addressing physician barriers to utilization may prove useful [35]. Additionally, understanding how physicians interact with clinical recommendations is an important aspect of examining the utilization of SDD. To be effectively implemented, practice recommendations must be specific about the population for which they are applied, as physicians are more likely to practice in a manner incongruent with a recommendation when the recommendation does not explicitly include a patient [36]. This is important in the context of hysterectomies in oncology, as the recommendations for SDD are written broadly for MIHs. Furthermore, non-adherence to recommendations is shown to be more frequent when making medical decisions in cancer, the elderly, and medically complex patients [36]. The younger age of the physician, physicians’ familiarity with a recommendation, clarity of physicians’ role in implementation, systemic support and facilitation, and physicians’ feeling of self-efficacy are all related to increased adherence to clinical recommendations [37].

### 4.3. Strengths and Limitations

The strength of these investigations is that the qualitative design captures nuanced ideas that are difficult to uncover in patient charts. Anecdotal accounts and summarized impressions, while not testable for statistical significance, yield important insights into implicit concepts not previously discussed. This approach helps to elucidate possible reasons for the lower utilization of the SDD amongst gynecologic oncologists that were observed in previous quantitative investigations. In addition, understanding how physicians view the recommendation of SDD provides an understanding of how to best implement practice change among physicians, as well as direct future quantitative investigation into the perceived barriers suggested by this qualitative work.

Limitations of this work include the small number of physicians available to be interviewed, focusing on physicians from one healthcare system, and the focus on a relatively narrow group of patients. While the relatively small number of physicians available to interview within this department is a potential limitation, their individual experiences span multiple practice sites by the nature of the infrastructure of the Division of Gynecologic Oncology at UPMC. While SDD recommendations themselves are not novel, investigating factors of adherence to these recommendations in the gynecologic oncology setting is novel. Some barriers identified in interviews are likely in flux as patients, physicians, and hospitals adapt to the guideline. The limited sample size of these investigations and the nature of qualitative research limits hypothesis testing and lends subjective impressions, but the thematic scheme revealed through the interviews provides a framework for further hypothesis-driven investigation. Despite the specificity of this study to hysterectomies, understanding non-health-related barriers to efficient and quality post-operative care is crucial across medical specialties.

### 4.4. Implications for Policy, Practice, and Research

In order to further explore the themes revealed in this study, quantitative analysis of the frequency with which admission is due to the presence of non-medical factors, such as patient distance from home, is needed. An idea of the relative frequency in which these factors affect care is necessary to eliminate the existing barriers to SDD within the oncologic population. Physician perception of patient discomfort with SDD supports the need for further qualitative research to assess patient perspectives regarding the practice of SDD after MIH within gynecologic oncology. While our group has previously qualitatively assessed the patient perception of the decision-making pathway between TAH and MIH, further qualitative interviewing at the time of discharge could elicit more subtle aspects of readiness for SDD [3]. Other potential avenues of research include the institution of a scheduling tool to optimize the timing of surgeries and resultant discharges. Additionally, future studies need to look more closely on discharge practices of patients with malignant vs. benign indications.

## 5. Conclusions

Physicians perceive that the utilization of SDD after an MIH in the gynecologic oncology patient population is limited by OSA, patients’ social support and access to transportation home, patients’ distance to travel post-operatively, inability to optimize surgical risk factors in the oncology population, and differences in the adoption of the SDD recommendation at different hospitals within the health system. Addressing a physician’s perceived barriers to SDD and catering recommendations specifically to the gynecologic oncology population may increase utilization. Multiple stakeholders need to be involved to improve processes related to the implementation of best practices [38].

## Figures and Tables

**Figure 1 jpm-12-01082-f001:**
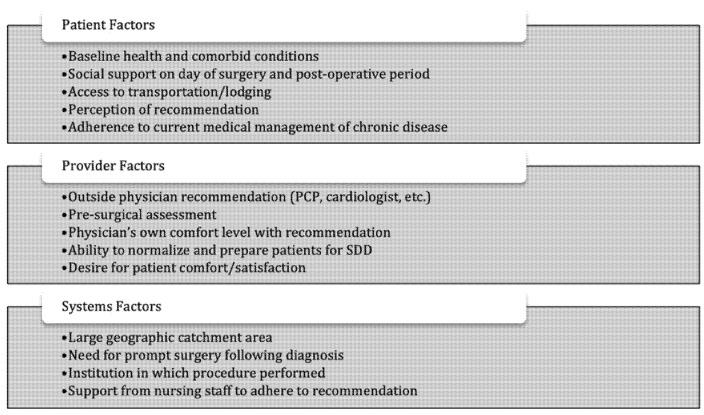
The barriers to SDD identified by the interviewed physicians fall into three major categories: patient factors, provider factors, and system factors.

**Table 1 jpm-12-01082-t001:** Example Interview Questions.

Question
Can you describe the pre-surgical counseling on same-day discharge patients receive before undergoing minimally invasive hysterectomy? What preparation beyond your conversation do they receive?
Why would you plan from the outset to keep a patient overnight following a minimally invasive hysterectomy? At what point is this decision made?
What would make you change your plan of course (admission to same-day discharge or same-day discharge to admission)?
Do you find the goal of same-day discharge within the division of Gynecologic Oncology feasible?
Do you feel that your division/department effectively encourages/supports utilizing same-day discharge?

**Table 2 jpm-12-01082-t002:** Obstructive Sleep Apnea and Unplanned Admission.

Interview Quotations
Particularly in our patients who have something like obstructive sleep apnea or obstructive sleep apnea that has not been diagnosed, and despite four to six hours in recovery… and respiratory therapy and breathing treatment, they just can’t safely wean them off oxygen. I can’t send them home with a new oxygen requirement that we have not worked up… we take them off oxygen and they have an oxygen saturation of 80%. They require additional workup.
Sleep apnea keeps patients—they have a hard time oxygenating in the post-op period.
If I cannot get a patient oxygenated, usually in someone who has undiagnosed sleep apnea, they have to be properly worked up and get their O2 up before we can safely send them home.
We try to schedule patients with obstructive sleep apnea earlier, so they have a little more time to wake up.
If someone has sleep apnea, I just tell them to make sure to bring their CPAP with them to the hospital… The bigger problem is patients who have been diagnosed but have thrown away their CPAP years ago. And you need to be well titrated on the positive pressure therapy before it is going to have a change on their wakefulness and their ability to respond to anesthesia.

**Table 3 jpm-12-01082-t003:** Surgical candidacy and medical comorbidities in oncology patients.

Interview Quotations
And the problem is… that you don’t have time to optimize anything because you are dealing with cancer patients. So it’s not like you have somebody who is morbidly obese who has dysfunctional uterine bleeding that you can try medical management, work with them—diet, exercise, all of those things—and [say] so you failed all of these other things, let’s take you to the OR in six to twelve months, right? You have just a small window and you do what you can to optimize them, but most of the time, you know these women all have diabetes, hypertension, morbid obesity, obstructive sleep apnea, and you just have to do what you can because you know you are going to have them in the OR within a month.
I think that from a post-operative issues or potential for issues standpoint, I think our patients probably have some of the highest risks for [issues] because we are operating on patients who, if they were going to have a truly elective surgery, they would not be a surgical candidate. But because they have a cancer diagnosis, we operate on them.
The [urogynecology] population is also an older population, but as that tends to be elective surgery, they don’t tend to have as many medical comorbidities.

**Table 4 jpm-12-01082-t004:** Site differences in utilization and execution of SDD.

Interview Quotations
At a place… that does mostly outpatient surgery and is set up to function that way, their same day discharge rates have always been way higher than everybody else, and the time that it takes them to get those patients out is much lower than the time that it takes [urban tertiary care hospital] or [suburban hospital] to get them out. At [the suburban outpatient hospital], my patients are routinely discharged in two hours of surgery, and [at the urban hospital], it’s more like four hours or longer.
I will say that the discharge process is different at [the suburban inpatient hospital] than at [the urban hospital]. There definitely has been a different roll out. At first, when we implemented the same-day discharge as part of the [Enhanced Recovery After Surgery protocol (ERAS)], women would stay in the PACU [post-anesthesia care unit] until discharge. It is an issue because the nursing staff is not as committed to the goal, and that gets transmitted to the patients. But now there is this ‘short stay’ ward, and patients lay down flat in bed, they get dinner, and the nurses ask them, ‘do you want to go home.’ And when it is posed as a question, they usually do not want to. Especially if it is late in the day. Then we get calls that patients are going to stay overnight. And I think it comes from the nurses thinking they are advocating, because the doctors are kicking the patients out, not that we are using evidenced-based medicine. And it’s the same ERAS, between [the urban hospital] and there, but the implementation has been different.
I think that the biggest hang-up in implementation was the PACU nursing discomfort with… getting used to what that looks like to send a patient home that quickly after a major abdominal operation. And that is very institution-dependent… when we started at [urban hospital] doing this, there was a lot of pushback and most of the patients that stayed didn’t stay because we wanted them to stay, they stayed because the nurses wanted them to stay. And that’s changed overtime as people have gotten more comfortable.

**Table 5 jpm-12-01082-t005:** Travel concerns and preoperative planning.

Interview Quotations
A lot of our patients come from four or five hours away, and so I will say... there is lodging in the city, there’s also a Family House ^a^ resource. But if that is something they can’t also afford, or if they don’t have anyone who can take them home, that is one other scenario where I have kept a patient overnight, but that is few and far between. If a woman doesn’t have family that day and can’t afford Family House, I’ll have her stay. If that makes our rate [of SDD go down], then I am okay with that. Her insurance will cover it.
I think with preoperative counseling, it is much more successful as well as I think much more accepted by the patient. Cause they have to plan around what they are going to do, you know? They have to have somebody to take them home. Then when I tell them that the vast majority of patients go home the same day, then it’s much better acceptance of that.
If somebody lives incredibly far I way, and I do have quite a few of those, I talk to them about options for staying at a hotel the night before, maybe the night after, or at Family House. But depending on resources, some women are unable to do that.
I will say, that sometimes, less so the patients, the families think it is crazy. To be discharged to Family House or a hotel at midnight. Or to drive home at midnight. They don’t understand why, even if we already talked about that as the plan… And I can’t really blame them. That is a long day, and we are sometimes asking a lot of people.
The other thing that factors in that shouldn’t but it does, is that we are being told these people have to go home, but we are also being judged on our patient satisfaction. They’ll say, I called and my insurance company says I am covered for an overnight admission. We push them out the door, but then we get dinged when it comes back that the patient satisfaction scores are not good.
Family House is a community resource available to patients and families needing lodging at a reduced rate while receiving medical care. It is typically utilized by patients and families who travel long distances to get care.

## Data Availability

Full transcripts and analyzed interviews available at request of the corresponding author, SB.

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
