# Peer review of "Qualitative Interview Study of Gynecologic Oncologist Utilization of Recommended Same-Day Discharge Following Minimally Invasive Hysterectomy"

_jpm, 2022, doi:10.3390/jpm12071082_

Round 1

Reviewer 1 Report

Although I think the subject is interesting I think the number of interviewed physicians is really very low, only 6 and, if I understood correct, from a single center. Which is an important limitation. I think including patients' interviews on preparedness for SDD or how they felt might be more valuable than responses by the doctor (table 4 and 5) and/or including interviews from different centers should be considered. 

I would like to have information on the average demographic parameters of the patients undergoing MIH in these centers (e.g. BMI, age) and the oncologic indications for MIH with SDD (was it only hysterectomy?). 

Furthermore (line 73) the location and type of hospital should be explained (it only says UPMC)

Reviewer 2 Report

  1. The similarity rate is 9%, which is acceptable.
  2. There is neither detailed method nor result of this study. 

Reviewer 3 Report

Dear authors ,

A study titled “ Qualitative interview study of gynecologic oncologist utilization of recommended same-day discharge following minimally 3 invasive hysterectomy” is presented.

Authors were aimed to  evaluate physician decision-making regarding same-day-discharge after minimally invasive hysterectomy for malignant indications.

Therefore, six physicians from The Division of Gynecologic Oncology within the Department of Obstetrics, Gynecology, and Reproductive Sciences, UPMC were interviewed and their answers were analyzed.

Barriers for considering same-day discharge were: obstructive sleep apnea (OSA), comorbidities, hospital site difference by doctors, travel time to hospital.

Despite a limit doctors interviewed, this article present a novel approach to evaluate quality aspects regarding early discharge. The study is well written and it is clear in its conclusions.

However, in order to confirm those observations, I would suggest to analyze also “quantitatively” those barriers. Would be interesting to evaluate: % of OSA, Charlson comorbidity Index, mean distance of patient’s home and hospital … for candidates to SDD (group 1) vs no-candidates for SDD (group 2) in those patients who underwent hysterectomy for malignancy during “a period of time”.

Furthermore, I suggest to review:

-Line 73: there is written as first time “UPMC”. I recommend to explain it.

-Line 185: “the” it appears at the end of line.

Thank You

Round 2

Reviewer 1 Report

Although I think it's very useful to explore why patients treated for malignancy are less likely to be discharged on the day of surgery, I think the setup of this study with physicians from 1 group in 1 area without including interviews with surgeons from other institutions or patients is a limitation, or compare to surgeons operating the same type of patients for elective, non-malignant indications

Reviewer 2 Report

1. Originality is the priority in consideration of a paper publication. During the plagiarism check, I found that the similarity rate is only 11%.

2. However, it is weird that there is no statistical method and results in this study. I have searched some published papers on the website with a similar topic, and all of them are supported by detailed statistical methods and results. From this point of view, I don’t think this paper is suitable for acceptance.
